# TTFields Prolonged the PFS of Epithelioid Glioblastoma Patient: A Case Report

**DOI:** 10.3390/brainsci13040633

**Published:** 2023-04-07

**Authors:** Yuxuan Ding, Qiang Wang, Feijiang Wang, Nan Wu, Jianrui Li, Xia He, Hao Pan, Lijun Wang

**Affiliations:** 1The Fourth School of Clinical Medicine, Nanjing Medical University, Nanjing 211166, China; 2Department of Neurosurgery, Jinling Hospital, Nanjing 210002, China; 3Department of Radiotherapy, The Affiliated Cancer Hospital of Nanjing Medical University, Jiangsu Cancer Hospital, Jiangsu Institute of Cancer Research, Nanjing 210009, China; 4Department of Pathology, Jinling Hospital, Nanjing 210002, China; 5Department of Diagnostic Radiology, Jinling Hospital, Nanjing 210002, China

**Keywords:** glioblastoma, case report, epithelioid glioblastoma, tumor-treating fields (TTFields), genetic diagnosis, BRAF V600E

## Abstract

Epithelioid glioblastoma (EGBM, classified as glioblastoma, IDH wild type, grade 4 according to the fifth edition of the World Health Organization (WHO) Classification of Tumors of the Central Nervous System (CNS) (WHO CNS5)) is a highly aggressive malignancy, with a median progression-free survival (mPFS) of about 6 months in adults. The application of tumor-treating fields (TTFields, possessing anti-cancer capabilities via anti-mitotic effects) in the maintenance of temozolomide (TMZ) chemotherapy showed a benefit for prolonging the mPFS of newly diagnosed glioblastoma (GBM) for patients for up to 6.9 months in the EF-14 clinical trial (NCT00916409). However, studies focusing on the effect of TTFields in EGBM treatment are very limited due to the rarity of EGBM. Here, we have reported a case of a 28-year-old male (recurrent left-sided limb twitching for 1 month and dizziness for 1 week) diagnosed with EGBM. A right frontal lobe occupancy was detected by magnetic resonance imaging (MRI), and a total tumor resection was performed. Meanwhile, a postoperative histopathology test, including immunohistochemistry and molecular characterization, was conducted, and the results revealed a BRAF V600E mutation, no co-deletion of 1p and 19q, and negative O-6-methylguanine DNA methyltransferase (MGMT) promoter methylation. Then, chemoradiotherapy was conducted, and TTFields and TMZ were performed sequentially. Notably, a long-term PFS of 34 months and a Karnofsky Performance Scale (KPS) of 90 were achieved by the patient on TTFields combined with TMZ, whose average daily usage of TTFields was higher than 90%.

## 1. Introduction

Epithelioid glioblastoma (EGBM) is an extremely rare neurological tumor, with fewer than 300 cases reported worldwide since the first case was discovered by Kepes in 1982, accounting for approximately 3% of all glioblastomas (GBM) [1,2,3,4]. Patients usually have a BRAFV600E mutation and isocitrate dehydrogenase-1 (IDH-1) wild type [5]. The treatment regimen for EGBM includes a surgical resection followed by adjuvant chemotherapy and radiation (the standard STUPP protocol). However, the median survival of treated adult patients is less than one year [5]. Several trials have demonstrated that temozolomide (TMZ) combined with tumor-treating fields (TTfields) prolongs progression-free survival in GBM patients [6,7]. Here, we report a case of a patient who was treated with early postoperative chemotherapy, TTFields, and long-term administration of TMZ. Until now, the patient has had a progression-free survival (PFS) of 34 months with a Karnofsky Performance Scale (KPS) score of 90. Compared to the E-14 trial, we used TMZ earlier (starting two weeks after surgery instead of week four), and in addition, we extended the duration of TMZ maintenance therapy, not for six cycles but for twelve cycles, and changed the dosing regimen for the last six cycles (75 mg/m^2^/d every eight weeks). To our knowledge, this is the first reported case of a patient treated with the above methods who survived for a long time. The relevant literature has been reviewed, and the diagnosis, especially regarding treatment strategy, has been discussed.

## 2. Case Presentation

On 14 November 2019, a 28-year-old male presented with a 1-month history of recurrent twitching of the left upper limb and 1 week of dizziness. Craniocerebral magnetic resonance imaging (MRI) showed space-occupying lesions in the right cingulate gyrus (Figure 1A–C). Five days later, the patient underwent a craniotomy for total tumor resection. The tumor tissue was seen intraoperatively as a cystic solid mass with abundant blood flow, and an unclear boundary with the surrounding brain tissue (Figure 2A–C).

Postoperative pathology showed, microscopically, that the tumor cells were significantly denser, interspersed with pseudo-fenestrated necrosis and microvascular hyperplasia. The EGBMs are dominated by a population of epithelioid cells with an abundance of eosinophilic cytoplasm, distinct cellular membranes, and a lack of cytoplasmic stellate processes (Figure 2D–F). Immunohistochemistry (IHC) demonstrated the following: BRAF V600E mutation, oligodendrocyte lineage transcription factor-2 (Olig-2) scattered cells positive, glial fibrillary acidic protein (GFAP) partial positive, Vimentin and S-100 positive, Integrase interactor 1 (INI1) positive, Brahma-related gene 1 (BRG1) positive, P53 positive, Ki-67 40% positive, Smooth Muscle Actin (SMA), and human melanoma black-45 (HMB-45) negative. Genetic testing showed the following: the O6-methylguanine-DNA-methyltransferase (MGMT) promoter was not methylated; 1p/19q genes were intact; there were no mutations in isocitrate dehydrogenase (IDH), or α-thalassemia/mental retardation syndrome X-linked gene (ATRX), or TP53; the telomerase reverse transcriptase promoter (TERTp); and the BRAF mutant. The IHC and genetic test results are summarized in Table 1 and Figure 3. The final histopathological diagnosis was epithelioid glioblastoma, IDH wild type, and WHO grade IV. The MRI was repeated 48 h after surgery and showed no obvious tumor residue. (Figure 1D–F). At this point, the patient’s left limb muscle strength was grade 3, with a KPS of 70.

In the fourth week after the surgery, the radiotherapist performed image-guided radio therapy (IGRT) on the tumor bed. The total radiation dose to the brain was 60 Gy, which was divided into 30 fractions (2.0 Gy per day, 5 consecutive days per week). Patients received oral TMZ daily from the second week after the end of surgery, administered on a body surface area basis (75 mg/m^2^/d) until the end of radiotherapy. After the completion of the concurrent chemoradiation, the patient then received 12 cycles of maintenance treatment with TMZ (the first cycle was 150 mg/m^2^/day, then 200 mg/m^2^/day for 5 days, per cycle of 28 days). TMZ doses were adjusted in February 2021 (75 mg/m^2^/d every 8 weeks) and discontinued in August 2022. Considering that the patient had a BRAF V600E mutation, we used the targeted drug vemurafenib during and after radiotherapy, but the patient discontinued it in each case because of severe nausea and vomiting, skin rash, and joint pain. After Multi-Disciplinary Treatment (MDT) and consultation with the family, the patient was started on the TTFields treatment on the 29 April 2020. The patient’s compliance with the treatment was good; he wore it for an average of more than 18 h per day, with only minor adverse effects during this period.

We conducted a long-term observation and follow-up of the patients, including imaging using the Response Evaluation Criteria in Neuro-Oncology (RANO) and evaluation of tumor response by an independent radiologist every 3 months according to RECISIT (The Response Evaluation Criteria in Solid Tumors) v1. In January 2020, a 2-month postoperative cranial MRI showed complete remission, and this good status continued until the last imaging examination in January 2023. The muscle strength of the patient’s left limb was grade 3 on the second day after surgery, and then it kept showing improvement, showing grade 4 four weeks after surgery and returning to normal muscle strength after 23 weeks. Adverse events (AEs) were collected from the time that the patients received treatment until the follow-up date. AEs were graded according to Common Terminology Criteria for Adverse Events (CTCAE), version 5.0, and the investigator assessed their causality. At post-operative week 5, the patient was treated with BRAF V600E inhibition, but stopped taking the drug in April 2020 due to intolerable arthralgia and rash. What is consistent with most reports is that TTFields has few adverse effects, it does not increase the risk of epilepsy, and does not produce hematological toxicity. By the third month of wearing, two rashes had occurred on the forehead and side of the head, but the rash healed after one month with medicine and adjusting the wearing period. We show in Figure 4 the picture of the patient’s head rash at the beginning and after treatment. As of the follow-up date (9 October 2022), a long-term PFS of 34 months (from the day of surgery to deadline of follow-up) and a KPS of 90 were achieved for the patient. We show a timeline with the relevant data from the episode of care in Figure 5.

## 3. Discussion

Epithelioid glioblastoma (EGBM) is a high-grade diffuse astrocytoma that occurs in children and young adults. It was abolished by the WHO Classification of Tumors of the Central Nervous System, 2021, and is now classified as a glioblastoma, IDH wild type [8]. EGBM is rapidly progressive and has a poor prognosis, with a median survival of 5 months for children and 6 months for adults [5]. The disease mainly occurs in the cerebral cortex, especially in the frontal lobe and temporal lobe [9]. The white matter collapse sign, meningeal tail sign, and encapsulation sign are characteristic changes in the disease that can be observed in an MRI. New MRI techniques, such as apparent diffusion coefficient (ADC) values, perfusion-weighted imaging (PWI), and Magnetic Resonance Spectroscopy (MRS), could be helpful for improving diagnostic accuracy [10,11]. These tumors can present a diagnostic challenge as they share an overlapping histopathological, genomic, as well as methylation profile with various other tumor types, particularly pleomorphic xanthoastrocytomas (PXAs) [4,12]. Tissue staining for EGBM reveals monotonous, closely packed large epithelioids and some rhabdoid cells with high mitotic activity, as well as microvascular proliferation and palisading necrosis [13]. Genetic testing shows BRAF mutations in more than 50% of patients, and EGBM cells commonly harbor TERTp mutations but lack both histone H3 and IDH mutations [1,14,15]. This case concerns a young male with a lack of specificity in clinical presentation and insufficient imaging findings to distinguish it from other malignant gliomas, which pose some challenges for an accurate clinical diagnosis. However, postoperative pathological immunohistochemistry and genetic testing can help clarify the diagnosis, and the results are basically consistent with the literature.

Because the incidence rate of EGBM is extremely low, its standard regimen follows the glioblastoma, or STUPP regimen, with complete surgical resection as possible, followed by concurrent chemoradiotherapy and six cycles of chemotherapy maintained with TMZ [9,16]. This standard Stupp regimen extends the overall survival (OS) from 12.1 months with postoperative radiotherapy alone to 14.6 months. However, the optimal duration of maintenance therapy remains a matter of debate. In clinical practice, investigators often prolong treatment in nonprogressive patients [17,18]. A secondary analysis of EORTC and NRG oncology/RTOG has explained that an adjuvant TMZ beyond six cycles is to some extent associated with a better PFS, especially in patients with methylated MGMTp [19]. Several retrospective studies also evaluate the safety and effectiveness of the long-term administration of adjuvant temozolomide [20,21,22]. In our report, the patient, who was MGMTp-unmethylated, began chemotherapy 14 days after surgery and was still on TMZ after six rounds, and he did receive survival benefits from this treatment regimen with no toxic side effects on regular hematology tests.

BRAF-V600E mutations are present in many tumors, including malignant melanoma (50%), papillary carcinoma of the thyroid (50–90%), lung cancer (3%), and colorectal cancer (5–10%). The development of BRAF inhibitors is recognized as an important therapeutic breakthrough for patients with malignant melanoma and papillary thyroid carcinoma [23]. In vitro studies confirmed that BRAFV600E inhibitors were effective in reducing the EGBM tumor cell viability and inhibiting the phosphorylation of key intracellular signaling [5]. Several clinical trials have supported the use of the BRAF inhibitor dabrafenib in pediatric patients with BRAF V600 mutation-positive low-grade gliomas and high-grade gliomas [24]. Patients from 13 countries with a positive BRAFV600E mutation received dabrafenib 150 mg twice daily plus trametinib 2 mg once daily orally. After approximately one year of follow-up, 15 of the 45 patients with high-grade glioma had an objective response [25]. There are several case reports that have highlighted the use of BRAFV600E inhibitors in patients with EGBM, and no specific adverse effects have been observed [26,27,28]. In the fifth postoperative week, the patient was treated with vemurafenib, which was discontinued after 3 months due to unacceptable side effects. We suspect that this side effect is related to the drug itself, as we tried vemurafenib after concurrent radiotherapy, which was not used in combination with any other treatment, and the patient still experienced intolerable side effects. We note that in the VE-BASKET study, investigators treated 24 BRAFV600-mutant patients with a regimen of vemurafenib 960 mg twice per day continuously, and during treatment, some patients also experienced adverse effects, including arthralgia, melanocytic nevus, and skin rash [23]. Additional prospective, randomized, and controlled clinical trials are urgently needed to confirm the optimal indication and drug safety of the drug for the treatment of EGBM.

The patient has been treated with the electric field until today, wearing it for more than 90% of the day. TTFields therapy uses alternating electric fields with a specific frequency (100~300 kHZ) to interfere with cell mitosis, and at the same time, the electrically charged material undergoes dielectrophoresis to rupture the cell membrane and kill the tumor. Furthermore, Kessler et al. showed that TTFields increase the permeability of the blood–brain barrier (BBB) by reducing the expression of tight junction proteins between neurovascular endothelial cells [29]. Kirson et al. combined TTFields with TMZ and found that TTFields could improve tumor sensitivity to chemotherapy [30]. In 2009, a phase 3 controlled trial (EF-14) was initiated, in which two-thirds of participants were randomized to receive TTFields (>18 h/day) plus adjuvant TMZ, while others received standard adjuvant TMZ maintenance therapy. The final report, published in 2017, showed that adding TTFields to TMZ maintenance therapy after chemoradiotherapy increased patient OS from 16.0 months with TMZ alone to 20.9 months and PFS from 4.0 to 6.7 months [6]. Most studies have demonstrated that the treatment is relatively safe because the alternating electric field only interferes with specific mitotic tumor cells and does not affect normal brain tissue [31,32]. In the phase 3 EF-11 trial for GBM, the TTFields group had significantly fewer adverse reactions of grade ≥ 2 (including gastrointestinal (4%), hematological (3%), and infectious (4%)) than those in the chemotherapy group (17%, 17%, and 8%), respectively. Severe AEs were significantly lower with TTFields versus chemotherapy (6% vs. 16%). In the EF-14 trial, adverse reactions occurred in 44% of patients receiving combination therapy, compared with 26% patients who used TMZ alone. Adverse reactions associated with TTField therapy are mainly rashes on the shaved scalp [16]. Based on the results of the EF-14 trial, in 2018, the National Comprehensive Cancer Network (NCCN) adopted “complete surgical excision plus concurrent chemoradiation plus TMZ in combination with TTFields” as a Class I recommendation for the treatment of new GBM. Genetic markers predicting treatment response were not elucidated. A retrospective study showed that in patients with glioma, people with PTEN mutations experienced a significant extension of PPS compared to PTEN wild-type patients [33]. At present, the use of TTFields in practical clinical practice is still rare (<12% in ndGBM and <16% in rGBM), especially in EGBM, where its safety and efficacy have not been fully elaborated [34]. We report a typical case of a patient with a glioblastoma epithelioid who had detailed genetic testing results and, after wearing daily for more than 90% of the time, obtained PFS for 34 months without serious adverse effects. Notably, a higher compliance with electric field therapy and better care of the scalp can lead to such good treatment results.

## 4. Conclusions

In conclusion, this is a rare case of a BRAFV600E mutation, MGMT promoter unmethylation glioblastoma patient, and the diagnosis mainly relies on pathological examination. The treatment regimen of early chemotherapy after total tumor resection, low-dose maintenance therapy of TMZ, and the addition of TTFields therapy resulted in a significant survival benefit for the patient. To our knowledge, this is the first reported case of a patient who was treated with the above methods and survived for a long time. Whether this treatment protocol can be extended to patients with BRAFV600E needs to be validated by more prospective studies, and we must continue to seek individualized and optimal treatment options for glioblastoma.

## Figures and Tables

**Figure 1 brainsci-13-00633-f001:**
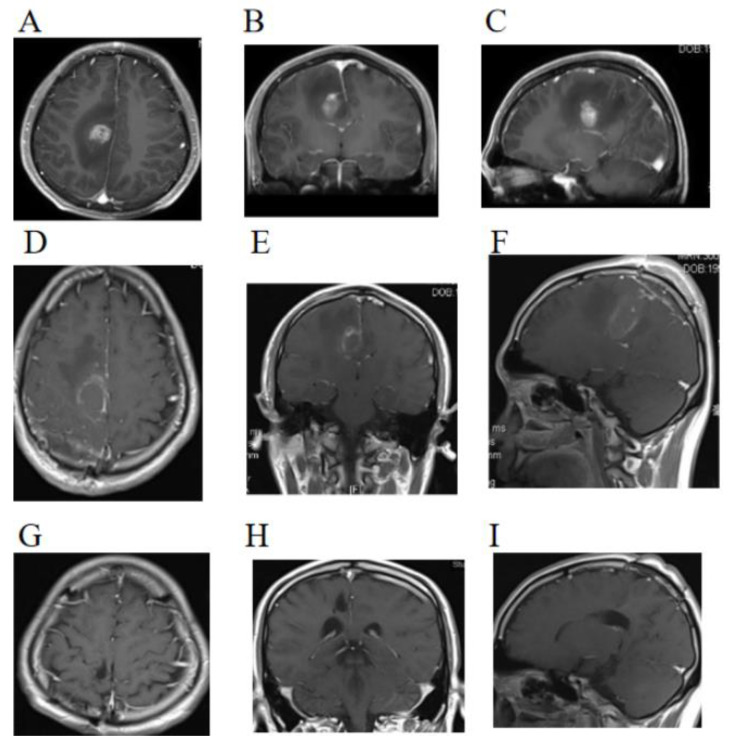
T1-weighted imaging of the brain. (**A**–**C**) Preoperative imaging revealed a mass in the right cingulate gyrus. (**D**–**F**) MRI 48 h after surgery showed no obvious tumor residue. (**G**–**I**) Images taken 21 months after surgery show complete remission of the lesion.

**Figure 2 brainsci-13-00633-f002:**
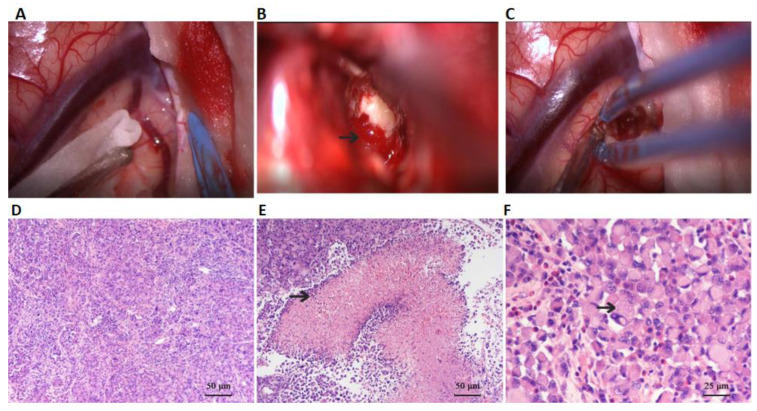
Intraoperative views and postoperative pathological specimens. (**A**,**C**) Some of the tumors are poorly defined. (**B**) The intraoperative visible tumor size is 20 mm × 10 mm × 20 mm. (**D**) Tumor cell density was significantly increased. (**E**) Extensive necrosis (arrow). (**F**) Plump epithelioid cells with laterally positioned nuclei and abundant cytoplasm (arrow).

**Figure 3 brainsci-13-00633-f003:**
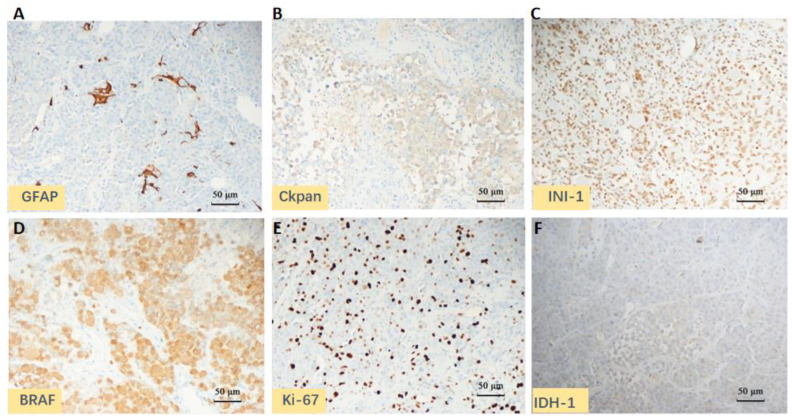
Immunohistochemical findings: (**A**) Immunohistochemical studies showed GFAP partial positive, and (**B**) Cytokeratin Pan (Ckpan) was partially positive. (**C**) Integrase interactor 1 (INI1) staining was universally intact. (**D**) Positive expression of BRAF V600E in EGBM. (**E**) Ki-67 was 40% positive. (**F**) IDH1 was negative.

**Figure 4 brainsci-13-00633-f004:**
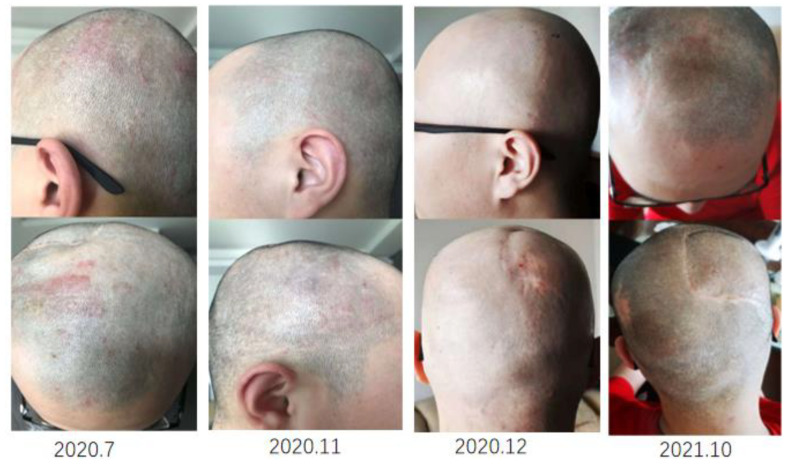
The picture of the patient’s head rash at the beginning and after treatment.

**Figure 5 brainsci-13-00633-f005:**
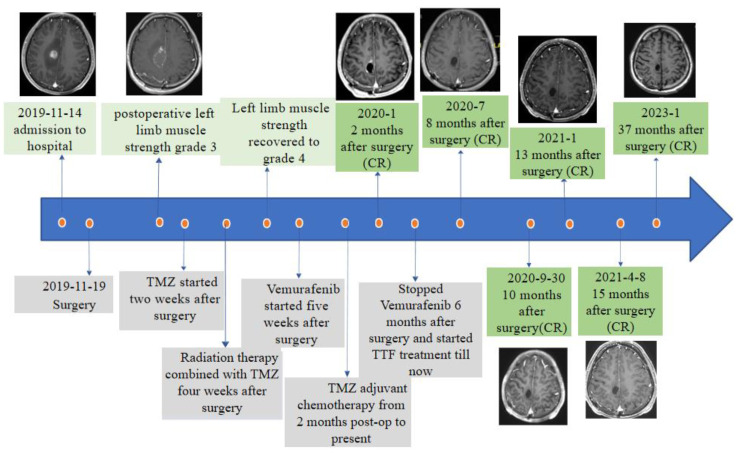
A timeline with relevant data from the episode of care.

**Table 1 brainsci-13-00633-t001:** Summary results of immunohistochemistry and genetic tests.

IHC	
Positive	Braf V600E, Ckpan (partial), GFAP (partial), Olig-2 (scattered), Vimentin, INI1, BRG1, S-100, Syn, P53, and Ki-67 (40%)
Negative	IDH1, CK7, CK20, Villin, H3-K27M, Neu-N, EMA, PR, SSTR2, PLAP, HMB-45, SMA, and Myogenin
**GENETIC TEST**	
1p/19q	intact
ATRX	not mutated
BRAFV600E	mutated
TERT	mutated
TP53	not mutated
MGMT promoter	not methylated

## Data Availability

The original contributions presented in the study are included in the article, further inquiries can be directed to the corresponding author.

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
