# Peer review of "TTFields Prolonged the PFS of Epithelioid Glioblastoma Patient: A Case Report"

_brainsci, 2023, doi:10.3390/brainsci13040633_

Round 1
Reviewer 1 Report
Authors report an extremely interesting case in which prolonged application of tumor treating fields (TTFs) improved significantly the median progression-free survival of a 28-year-old male affected by epithelioid glioblastoma (EGBM).
Due to the high recurrence and poor prognosis of GBM, together with the rarity of this specific type of GBM, the report is extremely valuable for the development of new therapeutic protocols. I appreciate Figure 4 which summarizes the patient's history. I would just require adding some interesting details and improving Figures and relative Legends.
Below are my specific comments:
1) Please highlight similarities and differences between the protocol adopted by the authors and the one(s) used in the mentioned clinical trials (lines 30-31 and lines 155-157). A scheme resuming the therapeutic protocol might help.
2) Please describe better the number and time distribution of TTF adverse effects. A comparison with the ones observed in clinical trials would be also valuable.
3) I am wondering if the adverse effects found by using vemurafenib might be due to the combination with the TTF application. Please comment on this.
Figures:
1) Please put magnification bars in Figures 1-3 and specify the dimensions of the tumor solid mass if possible.
2) Please specify better in Legend to Figure 1 the timing and the type of sections in the images.
3) Arrows could be used in Figure 2 to clearly indicate the elements described in lines 53-57.
4) Figure 3 misses Letters identifying single images. IDH-1 is not described in the Figure Legend and Ckpan is not specified.
Minor
1) Please report in the Text again the meaning of the abbreviations, as done in the Abstract, and add the missing ones (es: RECIST on line 94..lines 113-114……).
2) Please revise spacing
Reviewer 2 Report
In this manuscript by Ding Y at al., the authors have examined the effect of TTFields on an Epithelioid glioblastoma (EGBM) patient in conjunction with the TMZ. Such a treatment has shown a survival benefit in GBM patients. In this study the authors report the treatment benefit in an EGBM patients where they state that the patient is alive with a long term PFS of 34 months. This is a significant improvement in mPFS and treatment of more EGBM patients with TTFields will warrant the large benefit of treatment. Further, the genetic background of the patient, (in this case IDH WT, BRAF V600 mutant, 1p/9q intact and MGMT promoter non-methylated. It would be of significance to understand the genetic relevance of this genetic background in contributing to the long term PFS and hence further studies are required to warrant treatment.
Overall this is an important and relevant topic of study and warrants more investigation for the treatment of EGBM patients and also shed light on the current treatment of GBM patients. The paper has been written well and easy to understand, however minor grammatical corrections are required. The authors do address the main question posed, however, they should add to the relevance of the study by adding more information on TTFields in other GBM types specially in context to the different genetic makeup of the patients. I recommend a minor revision of this manuscript based on the following two comments.
1. What is the mPFS of an EGBM patient treated with the standard STUPP protocol, without TTFields. The authors should plot/compare PFS of the EGBM patient treated TTFields with other EGBM patients which did not receive TTFields.
2. They should also discuss the advantage of treating with TTFields in EGBM patients with and without the genetic mutations discussed in context of GBM patients treated with and without TTFields.
Reviewer 3 Report
The authors presented a very impressive case report regarding to the TTFields treatment approach for EGBM. From the scientific perspective, the manuscript was well composed. However, the language may need moderate improvement.
Author Response
Thank you very much for the suggestion, I have made some changes to the language and I will continue to improve if there are deficiencies.
Round 2
Reviewer 1 Report
The authors have addressed most of my comments and improved the manuscript, adding new info and a new Figure. I really appreciate this effort.
I would recommend in the future to better check the lines referred to by the reviewers by looking at the first version of the manuscript submitted and also to highlight the newly added lines in the revised version (there was also some discrepancy, e.g. in the reply to point 3, the new lines in the manuscript are 182-188 and not 203-211, as reported by the authors in their reply).
Below are the points still open:
1) I appreciate the answer to point 1, but I was referring to the clinical trials mentioned in lines 30-31 and 155-157 of version n.1 which are:
Lines 30-31 “Several trials have demonstrated that TMZ combined with TTfields prolongs progression-free survival in GBM patients (10–12)”.
Lines 155-157: “Professor Stupp led the EF-14 studies, which demonstrated that TTFields alone could improve quality of life in relapsed cases and prolong OS and PFS in new cases when combined with TMZ (37)”.
Please highlight similarities and differences between the protocol adopted by the authors and the one(s) used in the mentioned clinical trials (lines 30-31 and lines 155-157). A scheme resuming the therapeutic protocol might help.
Figures:
2) Figure 1, 2, and 3 present in version 2 of the manuscript still miss magnification bars. Please check and eventually add to all of them. Check also if the length represented by the magnification bar has to be added in the Figure Legend.
2) Maybe the arrows in Figure 2 are too big. Please reduce them and specify in the Legend what the arrow in Figure 2B indicates.
3) CKpan, IDH-1, and INI-1 abbreviations in Figure 3 Legend are still not specified
4) Probably Figures 4 and 5 should be inverted.
